# Nutritional Status Plays More Important Role in Determining Functional State in Older People Living in the Community than in Nursing Home Residents

**DOI:** 10.3390/nu12072042

**Published:** 2020-07-09

**Authors:** Małgorzata Pigłowska, Agnieszka Guligowska, Tomasz Kostka

**Affiliations:** Department of Geriatrics, Healthy Ageing Research Centre, Medical University of Lodz, 92-115 Lodz, Poland; agnieszka.guligowska@umed.lodz.pl (A.G.); tomasz.kostka@umed.lodz.pl (T.K.)

**Keywords:** nutritional status, malnutrition, functional status, older people, BIA

## Abstract

The aim of this study was to verify whether the relationship between nutritional and functional status differs between seniors in the community and those in long-term care institutions. One hundred nursing home (NH) residents aged 60 years and above and 100 sex- and age-matched community-dwelling (CD) older adults were examined. Functional status was assessed using the Comprehensive Geriatric Assessment (CGA) and nutritional status using anthropometric measures, the Mini Nutritional Assessment questionnaire (MNA) and bioimpedance analysis (BIA). Significant environmental interactions were observed with resting metabolic rate (RMR), extracellular water (ECW) and intracellular water (ICW) ratio, skeletal muscle mass (SMM), skeletal muscle index (SMI) and impedance (Z) and resistance (R) to the results of the Timed Up and Go (TUG) test. The two groups demonstrated different relationships between Z and R and handgrip strength and between Geriatric Depression Scale (GDS) score and fat free mass (FFM), body density, total body water (TBW) and phase angle (PhA). Nutritional status seems to be more important for functional state in CD older people than in NH residents. Therefore, to ensure the functional independence of seniors living in the community, it is crucial to maintain the correct nutritional parameters. Further studies are necessary to account for the fact that this relationship is less significant among NH residents and to identify other factors that may contribute to these discrepancies between community and institutional environments.

## 1. Introduction

Due to a wide range of physiological, psychological, social or environmental factors, older people are vulnerable to the development of nutritional impairments. Malnutrition, or undernutrition, is a frequent and often unrecognized problem in older adults [1,2]. Malnutrition can result in immune dysfunction, poor wound healing, anaemia, longer hospitalization and higher readmission rates, delayed recovery from surgery and a higher rate of mortality [3]. Various studies have associated malnutrition with poor functional status, impaired muscle strength, low bone mass and reduced cognitive function [3]. Beyond the individual burden, malnutrition also places an economic burden on the health care system by increasing healthcare costs [4].

The incidence of malnutrition varies between environments. Whereas the prevalence ranges between 5 and 10% among older adults living in the community, it may reach as high as 60% in institutionalized subjects [1,5,6]. Institutionalization is described as a factor that may be connected with poorer nutritional and functional status in older people [7,8], while living in a nursing home (NH) increases their degree of dependence [7,8,9,10]. In addition, a strong relationship has been found between malnutrition, and nutritional status in general, and functional status. However, although impaired functional status has been found to be related to impaired nutritional state among both NH residents [6] and community-dwelling (CD) older people [11,12], it remains unclear whether the two groups are equally susceptible to the impact of malnutrition. Therefore, the aim of the present study was to verify whether the relationship between nutritional and functional status differs between seniors in the community and those resident in long-term care institutions.

## 2. Materials and Methods

The study was conducted over a three-year period by the Department of Geriatrics, Medical University of Lodz, Poland. First, NH residents who met the inclusion criteria were selected and comprehensively examined. Following this, CD subjects were consecutively enrolled; each was sex- and age-matched (±1 year of age) to members of the NH group. All data was collected retrospectively and analysed statistically by experienced researchers from the Department of Geriatrics.

The inclusion criteria were age (60 years and above), ability to walk with or without auxiliary tools, verbal-logical contact and written informed consent to participate in the study. Subjects with a Mini Mental State Examination (MMSE) test score <11, who were not able to walk or in palliative conditions were excluded from the study. Due to the need for bioelectrical impedance analysis (BIA), pacemaker users and patients with oedema were also excluded. The study was approved by the Bioethics Committee of the Medical University of Łódź (Project identification code: RNN/863/11/KB) and complies with the Declaration of Helsinki and Good Clinical Practice Guidelines.

Multidimensional assessment, which included demographic and social parameters, health status, physical function, mental status, the risk of depression and nutritional status, was performed.

The functional status of the participants was assessed with scales and questionnaires included in the Comprehensive Geriatric Assessment (CGA). Basic and instrumental activities were examined by the Activities of Daily Living (ADL) Scale [13] and the Instrumental Activities of Daily Living (IADL) Scale [14]. The ADL scale contains six questions to assess the ability to perform basic activities (bathing, use of the toilet, continence, dressing, eating, mobility). The IADL scale includes eight components: using the phone, shopping, cooking, cleaning, washing, using means of transport, using medication and using money. The Timed Up and Go (TUG) test was performed to evaluate gait and balance [15]. During this test, the patient is asked to rise from a chair, walk three meters, turn around, walk back to the chair and sit down on the chair. A score above 14 s in this test indicates an increased risk of falling. Cognitive functions were evaluated with the Mini Mental State Examination (MMSE) test [16]; this includes six groups of tasks to assess basic psychological processes (orientation in time and place), memory, attention and counting, recall, language function and constructional praxis. Higher scores indicate better cognitive status. The maximum MMSE score is 30 points. A score of 24 to 26 points indicates mild cognitive decline without dementia, 19 to 23 suggests mild dementia, 11 to 18 moderate dementia and less than 11 indicates severe dementia.

The risk of depression was evaluated with the 15-item Geriatric Depression Scale (GDS) [17]. The tool contains 15 “yes” or “no” questions characterizing the depressive status of older people. The scores range from 0 to 15, with a higher score indicating deeper depression. A result of 0 to 5 points is considered normal, 6–10 points indicates a risk of depression and 11 points and more indicates depression. Muscle strength was assessed with the Jamar hydraulic hand dynamometer (model F12-0600) according to a standardized protocol [18]. Three measures of each handgrip were performed, and the maximum value was used in the analysis.

Nutritional status was evaluated using anthropometric measurements, the full version of the Mini Nutritional Assessment (MNA) [19] and via assessment of body composition. The anthropometric tests comprised body mass, body height, waist and calf circumference measurements and calculations of body mass index (BMI) and waist-to-height ratio (WHtR) [20]. The MNA questionnaire is the recommended test for detecting malnutrition and its risk in older people. It contains 18 questions, divided into screening and assessment parts. The total score ranges from 0 to 30, where a score of 24 or more points reflects normal nutrition, 17–23.5 points indicates risk of malnutrition and less than 17 points indicates malnutrition.

Body composition was assessed using bioimpedance analysis (BIA) with a Maltron BioScan 920-2, (Maltron International Ltd., Essex, UK) [21]. Whole body analysis was performed according to the manufacturer’s instructions: readings were taken using surface electrodes on the right side of the body; the participants were placed in a supine position, having completed a minimum six h fasting. The following parameters were obtained: fat mass (FM), fat-free mass (FFM), total body water (TBW) and body cell mass (BCM) expressed as percentage body mass. Extracellular water (ECW) and intracellular water (ICW) were shown as TBW components, and the ECW/ICW ratio was presented. The ratio of extracellular mass and body cell mass (ECM/BCM) was also given. Resting metabolic rate (RMR) was expressed in kilocalorie/kilogram (kcal/kg), body volume in litres (L) and body density in kg/L. Skeletal muscle mass (SMM) was obtained according to Janssen et al. [22] (presented in kg), and skeletal muscle index (SMI) was calculated and given in kg/m^2^ [23]. Moreover, the following raw data obtained from BIA were used: impedance (Z, Ohm), resistance (R, Ohm), reactance (Xc, Ohm) and the phase angle (PhA, degrees).

### Statistical Analysis

Statistica version 13 CSS software (StatSoft, Krakow, Poland) was used to perform statistical analysis. Data were verified for normality of distribution and equality of variance. The one-way analysis of variance (ANOVA) and Mann-Whitney test were used to compare the NH residents and community dwelling (CD) groups. A Chi^2^ test or Fisher’s exact test were used for comparisons between categorical variables. The results of the quantitative variables are presented as mean ± SD (standard deviation) for data with a normal distribution, and for data without a normal distribution they were additionally shown as median and quartiles. The correlations were assessed with Spearman correlation coefficients. To compare the lines of regression, logarithmic transformation was performed to achieve a normal distribution for the functional tests; however, these results are presented as standard values. The test of homogeneity of the slopes (i.e., comparison of regression lines and ANOVA interactions) were used to confirm the different relationship of nutritional and functional status according to the environment. Nutritional parameters were set as independent variables to verify the impact of environment on dependent variables (functional parameter). The limit of significance was assumed to be a *p*-value of 0.05 or less for all analyses.

## 3. Results

The study involved 200 older subjects aged 60 years and more (150 were women) who met the inclusion criteria. One hundred were NH residents recruited from three long-term care institutions, in Łódź (Poland): 37 out of 91 residents in one home, 27 subjects out of 100 residents in the second and 36 out of 195 residents in the third. Following this, community-dwelling sex- and age-matched outpatients of the Geriatrics Clinic were then consecutively enrolled in the study, one for each NH resident (±1 year of age).

The participants were diagnosed with the following concomitant diseases: arterial hypertension (NH = 67/CD = 63), history of myocardial infarction (NH = 11/CD = 9), ischemic heart disease (NH = 56/CD = 36), chronic heart failure (NH = 56/CD = 41), hypercholesterolemia (NH = 38/CD = 63), diabetes (NH = 25, CD = 24), history of stroke (NH = 20/CD = 13), osteoarthritis (NH = 56/CD = 57), osteoporosis (NH = 25/CD = 38), chronic obstructive pulmonary disease (NH = 20/CD = 13), eye diseases (NH = 52/CD = 56), depression (NH = 32/CD = 24), gastrointestinal diseases (NH = 27/CD = 40), urinary incontinence (NH = 40/CD = 37), faecal incontinence (NH = 7/CD = 8) and cancer (NH = 11/CD = 11). Thirty subjects in NH and six subjects in CD were current smokers (*p* < 0.001).

Table 1 presents the age, functional state and handgrip strength of the two groups. NH residents were characterized with worse functional status, both with regard to the basic and instrumental activities of daily living. They generally took longer to perform the TUG test and presented worse cognitive functions: no cognitive problems were identified by 64% of the CD group but only 31% of the NH group. NH residents also tended to display a greater number of depressive symptoms.

A comparison of nutritional status between the two groups, i.e., NH and CD, is presented in Table 2. NH residents were characterized by lower body mass, lower calf circumference and lower BMI. CD subjects obtained significantly better results in the MNA: 73% were characterized with the proper nutritional level compared to only 23% of residents in NHs; in addition, 14% of the NH group were malnourished and 63% were at risk of malnutrition. In the CD group, only 2% indicated malnutrition, but 25% were regarded as being at risk. The CD group had significantly higher body volume and a higher ICW level, but a lower ECW, ECW/ICW ratio and ECM/BCM index than the NH group. In addition, the CD participants were characterized with higher SMM and SMI. The raw BIA data also indicate great differences in body composition between the two groups. The CD group also demonstrated significantly lower Z and R and much higher PhA.

Relationships between age and selected body composition indicators and functional tests are presented in Table 3, for the NH group, and Table 4, for the CD group. Many more relationships were found between nutritional and functional status in the CD group than the NH group. Of the anthropometric parameters, body mass most strongly correlated with functional tests in the CD group, and of the functional tests, handgrip strength was most strongly related to nutritional status in both environments. Although MNA scores were related to all tests of functional status in the CD group, only the ADL, TUG score and handgrip strength were related to MNA in the NH group. Among NH residents, body composition parameters were mainly correlated with handgrip strength and the MNA test.

Although both groups demonstrated similar relationships between many components of nutritional status and functional status which were independent of the environment (with no environmental interaction), several of these relationships differed between the institutionalized and domestic environments. Significant statistical environmental interactions in the impact of RMR, ECW/ICW, SMM, SMI, Z and R on the results of TUG were found. Z and R demonstrated environmental interaction with handgrip strength, while FFM, body density, TBW and PhA with GDS score.

Some of the existing relationships between nutritional status and functional tests with environmental interactions are presented in Figure 1 (handgrip strength), Figure 2 (TUG test) and Figure 3 (GDS).

## 4. Discussion

The present study is the first such paper to examine whether the relationship of nutritional state to functional status differs between seniors in the community and those in institutional environments. Institutionalization may influence both nutritional and functional status. When first moving into an institution, seniors very often develop a mechanism of behaviour adjustment that includes them stopping performing everyday activities that they had done previously. They may adopt a passive attitude that creates dependency [24]. A previous study of NH residents found a deterioration in instrumental and even basic activities of daily living over the course of their first year in the institution [7], while another reports that the probability of maintaining functional capacity in older people decreased during two years of institutionalization [8,9]. In a 22-year observational study, González-Colaço et al. [25] found that NH residents presented greater cognitive decline than CD subjects. Our present study, comparing the functional status of sex- and age-matched older people living in two different environments, found that NH residents demonstrated greater dependency regarding both the basic and instrumental activities of daily living; they were also characterised by a lower level of mental functioning, a higher frequency of depressive symptoms and lower muscle strength than the CD group.

The prevalence of malnutrition is known to vary according to the environment [1,5,6]. Our present findings confirm that a disparity exists between institutionalized subjects and those living in the community. Simple anthropometric measures are common indicators of nutritional status. The loss of body mass is one of the most important risk factors of malnutrition in older adults: subjects with body mass loss present higher risk of infections, depression and mortality [26]. Interestingly, although a BMI under 23 kg/m^2^ or even under 25 kg/m^2^ has been associated with a higher risk of mortality, only a modest increase in mortality was found to be associated with elevated BMI among obese subjects, or no such relationship was associated with high BMI values [27,28]. In the present study, the CD subjects demonstrated higher calf circumference, body mass and BMI, indicating a better nutritional status in this environment.

The differences between the two examined environments were also visible in many body composition indices. The CD participants presented higher SMM and SMI, indicating differences in skeletal muscle quantity, and a lower ECM/BCM index; BCM is the metabolically active part of FFM related to muscle quantity. Volpato et al. [29] found that BCM was a strong and independent risk factor for mortality in older nursing home residents. Our findings also indicate significant differences in ECW/ICW between the two environments, with a ratio of about 2:3 indicating proper fluid retention. The ECW/TBW, known as the “oedema index”, is considered as one of the indicators of nutritional status—malnutrition leads to acute inflammation, decline in albumin level and oedema. ECW/TBW has been previously found to be negatively associated with serum albumin or haemoglobin level, and among a group of severely-malnourished, critically-ill patients, this ratio was found to be significantly higher among those who died than among the survivors [30].

It has been suggested that raw BIA data (R, Xc, PhA) should be used in subjects with an abnormal hydration level to avoid disturbances in the obtained results. The present findings indicate significant differences in raw data obtained with BIA between the two environments: R (which is inversely related to tissue hydration) and Z (combination of R and Xc) were lower in the CD group. Previously, PhA was found to be a predictor of increased mortality in hospitalized older patients [31] and to be significantly lower in non-survivors of critically ill patients [30]. Several studies have shown that bioelectrical PhA is related to nutritional status and, therefore, may serve as an indicator of the nutritional state of older people [32]. Moreover, it is positively associated with the level of serum albumin [30,32]. In the present study, NH residents were characterized with significantly lower PhA, suggesting decreased cellular integrity and poorer nutritional status in this group.

Malnutrition may be both the reason and the consequence of functional decline and impaired muscle strength. It is also negatively related with cognitive function as well as the risk of depression in older subjects [3,11,33,34]. Strong associations between nutritional and functional status have been observed in seniors in different environments. It has been found that among a group of patients of a day hospital with some limitations in basic and instrumental activities, the proportion of those demonstrating such problems significantly increased with a worsening of nutritional status [35]. A high prevalence of malnutrition (12%) and risk of malnutrition (57%) has been recorded among older people in the community (CD) receiving home care [36]; in addition, their functional status deteriorated significantly with a worsening of nutritional status.

Nutritional status improvement has previously been associated with better performance in the Short Physical Performance Battery (SPPB) and with a shorter time needed to perform the TUG test in older post-fall subjects [12]. Strong positive correlations have also been found between nutritional status and handgrip strength in older people [5,37]. Malnutrition was an independent determinant of poor functional status and low muscle strength in cancer patients [38]. In addition, a study of muscle strength and BIA in 223 healthy adults found that the raw BIA data differed significantly with regard to particular handgrip strength tertiles, suggesting that such measurements may be valuable in CD seniors [39].

Our findings indicate that nutritional status was clearly associated with the functional state of older people. However, these relationships were much more pronounced in seniors who lived in the community in comparison with NH residents. Of all the measured functional parameters, muscle strength was most strongly related to nutritional status in both environments. It is additionally worth mentioning that MNA was related to various functional tests, as well as with various nutritional parameters, including anthropometric and body composition measurements and BIA raw data; it is important to note that while all functional tests were related to MNA in the CD group, only muscle strength, ADL and TUG test were associated with MNA among NH residents. Hence, MNA appears to be a valuable tool for identifying subjects with poor body composition and may serve as an effective tool to allow screening for the functional decline of older adults. Our findings also indicate the presence a relationship between nutritional status and symptoms of depression, but only in the CD group. In contrast to some other reports [11] our study did not reveal any relationship between body mass, or BMI, and GDS, even in the CD group. This may be explained by previous data suggesting that this relationship may be nonlinear but U-shaped and by previous findings that both obesity and underweight are associated with the risk of depression [40].

Our findings also indicate that the environment has a significant influence on the relationship between particular elements of nutritional status and the results of the TUG, handgrip strength and GDS test. Such differences in this relationship between nutritional elements and functional status in these two environments merits further interest. However, this difference could be attributed to several potential factors. Firstly, NH residents presented significantly worse functional and nutritional levels, reflected in higher Z and R values and lower PhA scores than the CD group. Secondly, as Z and R are connected with body hydration, it is worth considering the role of water in preserving functional status and muscle strength. Some authors suggest that cellular hydration has a protective role against weakness, frailty status and functional decline [41]. A single bioimpedance spectroscopy (S-BIS) study of CD older subjects found the ECW/ICW ratio in the upper legs to be significantly inversely correlated with knee extension strength and gait speed, independent of age, sex, BMI and skeletal mass [42]. Serra-Prat et al. [43] found higher ICW to be associated with better functional level and lower risk of frailty of seniors from the community. These relationships were also significant in subjects with similar muscle mass but higher ICW [44]. These results highlight the protective effect of cell hydration in maintaining functional status, muscle performance and decreasing frailty risk in older subjects living in the community. Taking into account the elements of environmental interactions and based on the fact that water constitutes about ¾ of muscle mass, worse hydration status may be connected with lower muscle quality. Muscle quantity is strictly related to SMI and SMM, while muscle quality is related to lower myocyte hydration and may concern multiple factors, such as lower water intake or greater water loss, for example due to a reduced ability to concentrate urine. Our findings indicate similar interactions regarding ECW/ICW ratio, SMI and SMM.

Thirdly, physical activity (PA) level and usual activity level may also have an impact on various correlations in both environments. Increased PA exerts a number of influences on the overall health status of seniors, such as increasing PhA values and improving the integrity and functionality of cell walls [45]. In addition, a higher PA level, especially training that increases muscle mass, may lead to an increase in ICW and influence muscle quality [46]. Therefore, active seniors tend to display lower R and higher PhA [47,48,49]. Older people living in institutions are usually characterized by a lower level of PA in comparison with CD subjects [50]. What is more, institutionalization influences not only leisure PA but also usual daily living activities. Inactivity has detrimental effects on functional level, muscle strength, cognitive impairment and development of depression among seniors [51,52], and as such, the level of PA and its influence on analysed interactions needs to be explored.

Fourthly, in residents of long-term care facilities the presence of multiple chronic conditions in the same individual has profound implications for health status and may contribute to explaining the discrepancies in the relationship of nutritional state to functional status between community and institutional environments. Problems faced by chronically ill patients may include physical impairments, restrictions in daily activities, a negative body perception, decreased self-sufficiency and social stigmata [53,54,55]. Consequently, nutritional status seems to be more important for functional state in CD older people than in NH residents, in which a number of other factors influence functional state.

Fifthly, institutionalized seniors may be subject to other important factors, physical or psychological, connected with living in a NH that influence the development of depression, dementia or functional decline. Living in a NH and its associated social isolation both anticipate progress in dementia [56]. Isolation and loneliness are also associated with a rapid rate of motor decline [57]. It is also not insignificant that the NH residents presented poorer cognitive functions than the CD group: only 31% of them did not present any cognitive problems.

Numerous scientific reports highlight the possibility of improving nutritional state including muscle mass and functional fitness through nutritional supplementation and physical activity [58,59,60]. It has also been shown that a combination of physical exercise interventions with various nutritional approaches can modulate gut microbiota composition [61]. Sarcopenia and frailty syndrome are strongly associated with nutritional deficiencies, particularly a deficiency of protein in the diet [62]. The results obtained in the present study may be the starting point for planning future interventions, in which it will be possible to observe whether and how the introduced rehabilitation programmes and nutritional support will prevent atrophy, anabolic resistance and functional decline in older adults living in different environments.

Our study has some limitations. Many potential participants met the exclusion criteria, for example, the residents who were not able to stand were not included in the study. Our study did not include any biochemical parameters of nutritional status, such as serum albumin or haemoglobin level, that could reflect the nutritional state of our participants. In the study the BIA method was used for the assessment of body composition, which is not considered a ‘gold standard’; it was included because it is less demanding than other methods and easy to transport, which allowed body composition assessment to be performed for NH residents without requiring a visit to the clinic. Another limitation of our study is the fact that the PA level of participants was not examined. However, as PA influences functional level, muscle strength, cognitive status, risk of depression and body composition, it would be interesting to see whether it differed significantly between the two analysed groups. In addition, the rehabilitation programmes employed in the institutional environment varied, and this may also be a limitation of our study sampling method. Nevertheless, the main strengths of our study are connected with its use of sex- and age-matched groups and tests which assess different domains of functional level. In addition, it also analyses both body composition and raw data, which is not confounded by algorithms.

## 5. Conclusions

Our findings highlight significant discrepancies in the functional and nutritional status between the two tested groups. Of the two, the NH residents presented worse nutritional status. However, due to the high proportion of subjects at risk of malnutrition, nutritional status also needs to be improved in the general community. Significant environmental interactions were observed regarding the impact of particular elements of nutritional status on the results of TUG, muscle strength and GDS. Based on those environmental differences and interactions, nutritional status seems to play a more significant role regarding functional state in CD subjects. Therefore, it is necessary to maintain the correct nutritional parameters and perform systematic screening of nutritional status to prevent functional decline. Many factors may modify the influence of nutritional elements on functional status in the two environments, such as lower nutritional and functional status of NH residents, worse hydration, lower PA and the presence of comorbidities. Further studies are necessary to explain if and how other factors may contribute to these discrepancies between community and institutional environments.

## Figures and Tables

**Figure 1 nutrients-12-02042-f001:**
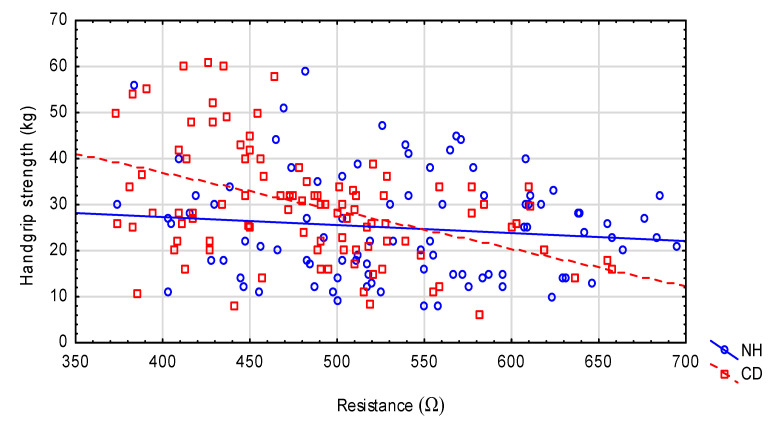
The relationship between handgrip strength and resistance in the NH and CD groups. NH: nursing home group; CD: community-dwelling group. Spearman’s rank correlation: NH (r = 0.021, *p* = 0.839); CD (r = −0.352, *p* = 0.0005). Interaction SMI* environment; *p* = 0.023.

**Figure 2 nutrients-12-02042-f002:**
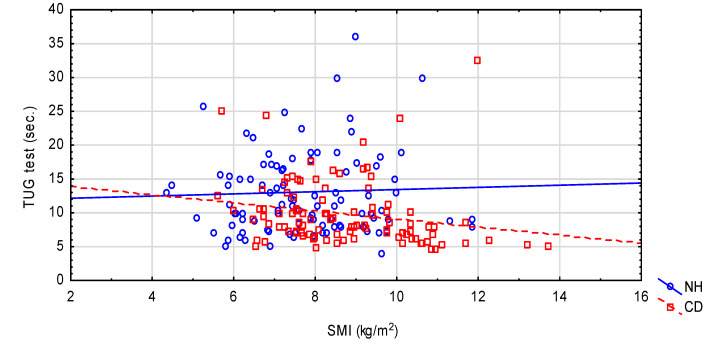
The relationship between TUG and SMI in the NH and CD groups. NH: nursing home group; CD: community-dwelling group. Spearman’s rank correlation: NH (r = 0.077, *p* = 0.443); CD (r = −0.334, *p* < 0.001). Interaction PhA* environment; *p* = 0.034.

**Figure 3 nutrients-12-02042-f003:**
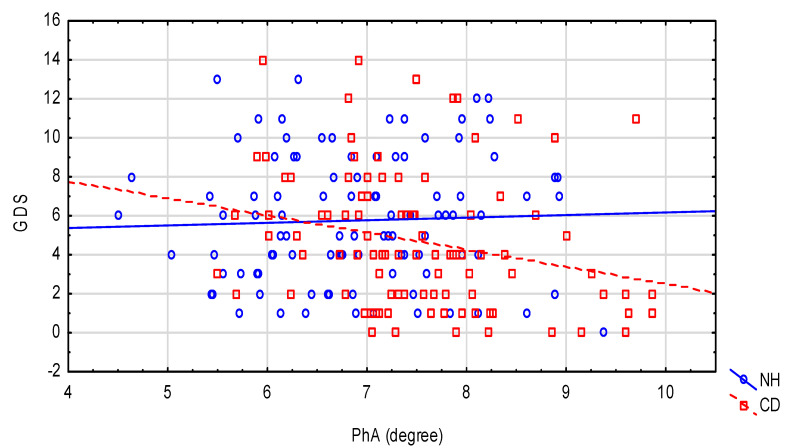
The relationship between GDS and PhA in the NH and CD groups. NH: nursing home group; CD: community-dwelling group.

**Table 1 nutrients-12-02042-t001:** A comparison of age, functional state and handgrip strength of the seniors in the nursing home (NH) and community-dwelling (CD) groups.

Variables	NH *n* = 100	CD *n* = 100	*p*-Value
Age (years)	74.6 ± 9.74 73 (67.0; 82.0)	74.9 ± 8.50 75 (66.5; 83.0)	NS
Women (%)	75	75	NS
ADL	5.54 ± 0.55 5.50 (5.50; 6.00)	5.73 ± 0.42 6.0 (5.50; 6.00)	0.015
IADL	4.65 ± 2.26 5.0 (3.0; 7.0)	7.13 ± 1.71 8.0 (7.0; 8.0)	<0.001
TUG	13.0 ± 7.07 10.98 (8.16; 16.38)	9.88 ± 4.89 8.3 (6.66; 11.0)	<0.001
MMSE	23.7 ± 4.69 24.0 (22.0; 27.0)	26.8 ± 3.84 28.0 (25.0; 30.0)	<0.001
GDS	5.76 ± 3.28 5.50 (3.00; 8.00)	4.65 ± 3.53 4.0 (2.0; 6.5)	0.01
Handgrip strength (kg)	24.7 ± 11.2 (23.0 15.0; 32.0)	30.2 ± 12.7 29.0 (21.5; 36.25)	0.002

ADL: Activities of Daily Living Scale; IADL: Instrumental Activities of Daily Living Scale; TUG: Timed Up and Go test; MMSE: Mini Mental State Examination questionnaire; GDS: Geriatric Depression Scale; NH: nursing home group; CD: community-dwelling group. The results for data with a normal distribution are presented as mean ± SD. The results for data without a normal distribution are presented as mean ± SD and additionally as median and quartiles.

**Table 2 nutrients-12-02042-t002:** A comparison of nutritional status of the seniors in the NH and CD groups.

Variable	NH *n* = 100	CD *n* = 100	*p*-Value
Body mass (kg)	67.1 ± 15.7 65.0 (56.55; 78.00)	72.6 ± 14.0 71.05 (63.45; 80.55)	0.014
BMI (kg/m^2^)	27.2 ± 5.66 26.7 (22.6; 31.7)	28.7± 4.65 28.2 (25.3; 30.6)	0.043
Waist circumference (cm)	94.3 ± 13.4	93.8 ±11.2	NS
WHtR	0.602 ± 0.088	0.591 ± 0.070	NS
Calf circumference (cm)	35.1 ± 3.90 34.7 (32.0; 37.0)	37.1 ± 4.04 37.0 (35.0; 39.0)	<0.001
MNA	21.1 ± 3.42 21.3 (18.5; 23.5)	25.4 ± 3.29 26.0 (23.5; 28.0)	<0.001
FM (%)	37.6 ± 10.2	37.1 ± 8.15	NS
FFM (%)	62.4 ± 10.2	62.9 ± 8.15	NS
TBW (%)	50.5 ± 6.28 50.0 (45.8; 53.3)	51.2 ± 5.41 50.9 (47.7; 54.4)	NS
ECW (%)	47.3 ± 2.50 47.6 (45.9; 48.8)	46.4 ± 2.13 47.1 (45.0; 47.9)	0.007
ICW (%)	52.7 ± 2.50 52.4 (51.2; 54.1)	53.6 ± 2.14 52.9 (52.1; 55.0)	0.008
ECW/ICW	0.90 ± 0.09 0.91 (0.85; 0.95)	0.87 ± 0.07 0.89 (0.82; 0.92)	0.007
BCM (%)	34.8 ± 5.26 34.0 (31.2; 37.5)	35.6 ± 4.72 34.6 (32.3; 38.2)	NS
ECM/BCM	0.80 ± 0.09 0.80 (0.75; 0.84)	0.77 ± 0.07 0.78 (0.75; 0.81)	0.037
RMR/kg (kcal/kg)	19.1 ± 3.57 18.8 (16.2; 21.3)	18.2 ± 2.42 18.0 (16.7; 20.0)	NS
Body volume (Lt)	66.5 ± 15.9 64.7 (55.2; 77.4)	71.3 ± 14.2 69.6 (62.0; 78.4)	0.030
Body density(kg/Lt)	1.02 ± 0.02 1.01 (1.00; 1.03)	1.02 ± 0.02 1.01 (1.00;1.03)	NS
SMM (kg)	19.4 ±5.77 18.4 (15.6; 22.9)	22.4 ± 6.14 21.1 (17.6; 25.4)	<0.001
SMI (kg/m^2^)	7.75 ± 1.52 7.60 (6.72; 8.80)	8.74 ± 1.64 8.44 (7.54; 9.93)	<0.001
Z (Ω)	552.7 ± 93.2	485.0 ± 69.6	<0.001
R (Ω)	549.0 ± 92.4	481.3 ± 69.4	<0.001
Xc (Ω)	66.7 ± 15.7 66.5 (56.3; 74.5)	63.5 ± 10.26 63.8 (55.7; 70.5)	NS
PhA (degrees)	6.90 ± 1.005	7.55 ± 1.03	<0.001

BMI: body mass index; WHtR: waist-to-height ratio; MNA: Mini Nutritional Assessment; FM: fat mass; FFM: fat-free mass; TBW: total body water; ECW: extracellular water; ICW: intracellular water; ECW/ICW: extracellular water and intracellular water ratio; BCM: body cell mass; ECM/BCM: extracellular mass and body cell mass ratio; RMR: resting metabolic rate; SMM: skeletal muscle mass; SMI: skeletal muscle index; Z: impedance; R: resistance; Xc: reactance; PhA: phase angle; Ω: Ohm; NH: nursing home group; CD: community-dwelling group. The results for data with a normal distribution are presented as mean ± SD. The results for data without a normal distribution are presented as mean ± SD and additionally as median and quartiles.

**Table 3 nutrients-12-02042-t003:** Relationships between age and selected indicators of nutritional status and body composition to functional tests in NH residents.

Variable	Handgrip Strength	ADL	IADL	TUG Test	MNA	MMSE	GDS
Age	−0.378 ***				−0.288 **		
Body mass (kg)					0.362 ***		
BMI (kg/m^2^)							
Waist circ. (cm)					0.243 *		
WHtR	−0.266 **						
Calf circ. (cm)					0.245 *		
MNA	0.292 **	0.272 **		−0.206 *			
FM (%)	−0.438***						
FFM (%)	0.438 ***						
TBW (%)	0.305 **				−0.219 *		
ECW (%)	−0.530 ***				−0.305 **		
ICW (%)	0.530***				0.305 **		
ECW/ICW	−0.535 ***				−0.303 **		
BCM (%)	0.421 ***						
ECM (%)	0.395 ***						
ECM/BCM							
RMR/kg (kcal/kg)	0.247 *						
Body volume (Lt)					0.333 ***		
Body density	0.438 ***						
SMM (kg)	0.533 ***				0.214 *		
SMI (kg/m^2^)	0.396 ***						
Z (Ω)							
R (Ω)							
Xc (Ω)	0.226 *						
PhA (degrees)	0.414 ***			−0.214	0.212 *		

* *p* < 0.05; ** *p* < 0.01; *** *p* < 0.001. BMI: body mass index; WHtR: waist-to-height ratio; MNA: Mini Nutritional Assessment; FM: fat mass; FFM: fat-free mass; TBW: total body water; ECW: extracellular water; ICW: intracellular water; ECW/ICW: extracellular water and intracellular water ratio; BCM: body cell mass; ECM/BCM: extracellular mass and body cell mass ratio; RMR: resting metabolic rate; SMM: skeletal muscle mass; SMI: skeletal muscle index; Z: impedance; R: resistance; Xc: reactance; PhA: phase angle; Ω: Ohm; ADL: Activities of Daily Living Scale; IADL: Instrumental Activities of Daily Living Scale; TUG: Timed Up and Go test; MMSE: Mini Mental State Examination questionnaire; GDS: Geriatric Depression Scale.

**Table 4 nutrients-12-02042-t004:** Relationships between age, and selected indicators of nutritional status and body composition, and functional tests in the CD group.

Variable	Handgrip Strength	ADL	IADL	TUG	MNA	MMSE	GDS
Age (years)	−0.438 ***		−0.238 *	0.481 ***	−0.207 *	−0.533 ***	0.201 *
Body mass (kg)	0.377 ***		0.0734	−0.202 *	0.1488	0.317 **	
BMI (kg/m^2^)							
Waist circ. (cm)							
WHtR							
Calf circ (cm)						0.213 *	
MNA	0.318 **	0.272 **	0.438 ***	−0.464 ***		0.364 ***	−0.527 ***
FM (%)	−0.540 ***	−0.231 *		0.322 **	−0.225 *		0.275 **
FFM (%)	0.540 ***	0.231 *		−0.322 **	0.225 *		−0.275 **
TBW (%)	0.420 ***	0.219 *		−0.229 *			−0.238 *
ECW (%)	−0.599 ***			0.341 ***	−0.245 *	−0.395 ***	0.204 *
ICW (%)	0.600 ***			−0.341 ***	0.246 *	0.396 ***	−0.204 *
ECW/ICW	−0.600 ***			0.344 ***	−0.246 *	−0.398 ***	0.205 *
BCM (%)	0.496 ***			−0.281 **			−0.226 *
ECM (%)	0.496 ***	0.199 *		−0.315 **			−0.258 *
ECM/BCM							
RMR/kg (kcal/kg)	0.321 **			−0.267 **			
Body volume (Lt)	0.322 **					0.317 **	
Body density (kcal/Lt)	0.540 ***	0.230 *		−0.326 **	0.228 *		−0.276 **
SMM (kg)	0.681 ***			−0.399 ***	0.285 **	0.329 **	−0.204 *
SMI (kg/m^2^)	0.605 ***			−0.345 **	0.262 **	0.266 **	−0.226 *
Z (Ω)	−0.395 ***			0.2154 *			
R (Ω)	−0.402 ***			0.220 *			
Xc (Ω)							
PhA (degrees)	0.457 ***		0.271 **	−0.453 ***	0.324 **	0.413 ***	−0.307 **

* *p* < 0.05; ** *p* < 0.01; *** *p* < 0.001. BMI: body mass index; WHtR: waist-to-height ratio; MNA: Mini Nutritional Assessment; FM: fat mass; FFM: fat-free mass; TBW: total body water; ECW: extracellular water; ICW: intracellular water; ECW/ICW: extracellular water and intracellular water ratio; BCM: body cell mass; ECM/BCM: extracellular mass and body cell mass ratio; RMR: resting metabolic rate; SMM: skeletal muscle mass; SMI: skeletal muscle index; Z: impedance; R: resistance; Xc: reactance; PhA: phase angle; Ω: Ohm; ADL: Activities of Daily Living Scale; IADL: Instrumental Activities of Daily Living Scale; TUG: Timed Up and Go test; MMSE: Mini Mental State Examination questionnaire; GDS: Geriatric Depression Scale. Spearman’s Rank Correlation: NH (r = −0.075, *p* = 0.458); CD (r = −0.399, *p* < 0.001). Interaction resistance * environment; *p* = 0.015.

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
