# Peer review of "Nutritional Status Plays More Important Role in Determining Functional State in Older People Living in the Community than in Nursing Home Residents"

_nutrients, 2020, doi:10.3390/nu12072042_

Round 1
Reviewer 1 Report
Dear Authors,
the topic is very interesting, and the manuscript is well written, although some issues should be clarified to improve the paper.
My major concern is that physical activity levels in both groups were not assessed. However, it could severely affect study results and the clinical impact of the study especially considering the outcomes evaluated. In addition, NH Group was composed of patients selected from NH and from Rehabilitation and Care Centre, but the potential differences in rehabilitation programs could have severe implications in functional state and consequently affect the homogeneity of the sample.
Given that you assessed musculoskeletal muscle mass, functional status, and muscle strength, why did you not considered to stratify the sample for sarcopenia?
The Nursing Home sample derives after the exclusion of 286 patients in a consecutive series of 386 patients during the study period. Do you think the high number of patients excluded could limit the potential generalization of these results to real-world elderly patients?
Furthermore, most developed world countries have accepted the chronological age of 65 years as a definition of 'elderly' or older person. Why did you include patients aged 60 years and above?
Lastly, Discussion Section should be improved highlighting the clinical impact of study results and potential therapeutic option contrasting age-related functional decline in elderly, and largely underline the limitations of the study taking into account the previous considerations.
Major revision
- MATERIALS AND METHODS: The NH sample derives after the exclusion of 286 patients in a consecutive series of 386 patients during the study period. Do you think the high number of patients excluded could limit the potential generalization of these results to real-world elderly patients?
- MATERIALS AND METHODS: Most developed world countries have accepted the chronological age of 65 years as a definition of 'elderly' or older person. Why did you include patients aged 60 years and above?
- MATERIALS AND METHODS: Given that you assessed musculoskeletal muscle mass, functional status, and muscle strength, why did you not considered to stratify the sample for sarcopenia?
- DISCUSSION: This Section should be improved highlighting the clinical impact of study results and potential therapeutic option, including physical exercise and nutritional supplementation in order to contrast the age-related functional decline in elderly, according to the recent scientific literature.
According to this, you could cite the following references:
- Marshall RN et al. Nutritional Strategies to Offset Disuse-Induced Skeletal Muscle Atrophy and Anabolic Resistance in Older Adults: From Whole-Foods to Isolated Ingredients. Nutrients. 2020;12(5):E1533. doi:10.3390/nu12051533.
- de Sire A et al. Myostatin as a potential biomarker to monitor sarcopenia in hip fracture patients undergoing a multidisciplinary rehabilitation and nutritional treatment: a preliminary study. Aging Clin Exp Res. 2020 May;32(5):959-962. doi: 10.1007/s40520-019-01436-8.
- Landi F, Calvani R, Tosato M, et al. Protein Intake and Muscle Health in Old Age: From Biological Plausibility to Clinical Evidence. 2016;8(5).pii:E295.
- de Sire A et al. Gut-Joint Axis: The Role of Physical Exercise on Gut Microbiota Modulation in Older People with Osteoarthritis. Nutrients. 2020 Feb 22;12(2). pii: E574. doi: 10.3390/nu12020574.
- de Sire A et al. Nutritional supplementation in hip fracture sarcopenic patients: A narrative review. Clin. Cases Miner. Bone Metab. 2019 Jan-Apr;16(1):27-30.
- DISCUSSION: Considering the outcomes assessed in the study, physical activity levels in both groups should have been screened because it could have severe implications not only in physical function but also in body composition and depression. Therefore, significative differences in physical activity levels between groups could affect study results and the clinical impact of the study. Please report this limitation in ‘Study limitations’ subsection.
- DISCUSSION: differences in rehabilitation programs (i.e time, intensity, numbers of sessions) among Nursing Homes and Rehabilitation and Care Centre composing NH Group could severely affect functional state and consequently homogeneity of the sample. Please report this possible sampling BIAS in ‘Study limitations’ subsection.
Minor revisions
- MATERIALS AND METHODS. Please report all data and numbers in “Results” section instead of “Materials and methods” Section.
- MATERIALS AND METHODS. This section should be improved clarifying the study period, data collection process, and study design, underlining if data were collected retrospectively or prospectively.
- MATERIALS AND METHODS. Please clarify who performed the analysis and eventually if it was blinded.
- TABLES: Please clarify all abbreviations in a legend at the end of the tables.
Reviewer 2 Report
I n this excekllent piece of work ,you highlighted the role of serum albumin and haeoglobin as parameters to assess nutritional status in addition to the anthropometric measurments which you have already performed.Since these were not measured in your study it would be useful to mention that as one of the limitations of the study in the Discussion.
Reviewer 3 Report
In view of the aging of Western societies, the topic of work is very important and necessary. The authors obtained interesting research results.
However, they concern a small group of people from one city in Poland, so you should agree with the authors that more research is needed.
The elderly, especially residents of long-term care facilities, constitute a particularly vulnerable group due to specific needs, including nutritional needs. They are characterized by functional disability, multiple morbidity, quantitative and / or qualitative malnutrition, which leads to adverse changes in all organs and body systems, deteriorating quality and life expectancy.
The authors in the discussion stated that there are no studies comparing the studied features of elderly people living in and outside nursing homes. I also have not found them. However, you can find research in this area, including in Poland.
Below are a few examples of interesting works from this area (for consideration by the authors).
- Ożga E., Małgorzewicz S. Nutritional assessment of the elderly. Geriatrics 2013; 7, 2, 98-103
- Schlegel-Zawadzka M. et al. Assessment of the ability of older people to self-service and self-care, including eating behavior. 21st Century Nursing 2011; 35, 2, 5-9
- StrugaÅ‚a M, Wieczorkowska-Tobis. Ocena stanu odżywienia pacjentów OddziaÅ‚u Geriatrycznego w kontekÅ›cie ich sprawnoÅ›ci funkcjonalnej. 2011;(5):89-93
- HumaÅ„ska MA, KÄ™dziora-Kornatowska K. WpÅ‚yw miejsca zamieszkania osób w podeszÅ‚ym wieku na stan odżywiania siÄ™. Gerontol Pol 2009;17(3):126-8
- Lengyel Christina O. et al. Nutrient Inadequacies Among Elderly Residents Of Long-term Care Facilities. Canadian Journal of Dietetic Practice and Research; Summer 2008; 69, 2, 82-87.
- Kulnik D, Elmadfa I. Assessment of the Nutritional Situation of Elderly Nursing Home Residents in Vienna Ann Nutr Metab 2008;52 (suppl 1):51–53.
- Sakineh Nouri Saeidlou, et al. Assessment of the nutritional status of elderly. People living at nursing homes in northwest Iran. International journal of academic research. Vol. 3. No. 1. January, 2011, Part II
- Kvamme JM, et al. Body mass index and mortality in elderly men and women: the Tromsø and HUNT studies. J Epidemiol Community Health 2012;66(7):611-7
Tables 1 and 2.
It is worth explaining for what refer data in the first and second row e.g.
|
NH |
CD |
|
|
Age (years) |
74.6 ± 9.74 73 (67.0; 82.0)
|
74.9 ± 8.50 75 (66.5; 83.0) |
Round 2
Reviewer 1 Report
The manuscript has been adequately revised.
Author Response
Dear Reviewer,
Thank you very much for your time and positive opinion about our paper.